# Evaluating Accuracy in Five Commercial Sleep-Tracking Devices Compared to Research-Grade Actigraphy and Polysomnography

**DOI:** 10.3390/s24020635

**Published:** 2024-01-19

**Authors:** Kyle A. Kainec, Jamie Caccavaro, Morgan Barnes, Chloe Hoff, Annika Berlin, Rebecca M. C. Spencer

**Affiliations:** 1Neuroscience & Behavior Program, French Hall, University of Massachusetts Amherst, 230 Stockbridge Road, Amherst, MA 01003, USA; kkainec@umass.edu; 2Institute for Applied Life Sciences, Life Science Laboratories, University of Massachusetts Amherst, 240 Thatcher Road, Amherst, MA 01003, USA; morganbarnes@umass.edu (M.B.); cehoff@umass.edu (C.H.);; 3Department of Psychological and Brain Sciences, Tobin Hall, University of Massachusetts Amherst, 135 Hicks Way, Amherst, MA 01003, USA

**Keywords:** Fitbit, Garmin, Oura, Withings, actigraphy, polysomnography, sleep

## Abstract

The development of consumer sleep-tracking technologies has outpaced the scientific evaluation of their accuracy. In this study, five consumer sleep-tracking devices, research-grade actigraphy, and polysomnography were used simultaneously to monitor the overnight sleep of fifty-three young adults in the lab for one night. Biases and limits of agreement were assessed to determine how sleep stage estimates for each device and research-grade actigraphy differed from polysomnography-derived measures. Every device, except the Garmin Vivosmart, was able to estimate total sleep time comparably to research-grade actigraphy. All devices overestimated nights with shorter wake times and underestimated nights with longer wake times. For light sleep, absolute bias was low for the Fitbit Inspire and Fitbit Versa. The Withings Mat and Garmin Vivosmart overestimated shorter light sleep and underestimated longer light sleep. The Oura Ring underestimated light sleep of any duration. For deep sleep, bias was low for the Withings Mat and Garmin Vivosmart while other devices overestimated shorter and underestimated longer times. For REM sleep, bias was low for all devices. Taken together, these results suggest that proportional bias patterns in consumer sleep-tracking technologies are prevalent and could have important implications for their overall accuracy.

## 1. Introduction

Consumer sleep-tracking technologies, smart devices that non-invasively monitor sleep, have become increasingly advanced, utilize more physiological signals, and claim to measure an increasing repertoire of sleep metrics more accurately [1,2,3,4]. Some commercial devices perform as good as research-grade actigraphy for detecting sleep [5,6], though their algorithms do not meet the established reporting standards for reproducibility [7,8,9,10]. Commercial devices could be a cost-effective and logistically advantageous alternative for the scientific and clinical evaluation of sleep. However, a lack of scientific evaluation remains a critical barrier to their clinical implementation [11,12].

Sleep is characterized by physiological changes in brain activity, heart rate, blood pressure, body temperature, respiration, eye movement, and muscle tone [13,14,15]. To characterize sleep most accurately, the gold standard measure, polysomnography (PSG), utilizes a montage of electrodes and other recording devices that capture these physiological changes [14,15,16,17]. Offline, trained experts apply standardized criteria to determine sleep stages in 30 s intervals [18]. Within each 30 s interval, scorers identify changes in physiological patterns to characterize sleep stages [18,19]. Although PSG is the gold standard for characterizing sleep stages, applying the PSG montage and identifying sleep stages requires hours of time from multiple trained professionals [16,17]. The inter-scorer reliability of even well-trained scorers ranges from 70 to 100% [16,17,20,21]. PSG requires costly equipment and is uncomfortable for participants [16,22,23]. While automated sleep staging algorithms have been developed, they do not reduce the cost, time, or discomfort of PSG [21,24,25]. Easier, comparably accurate, and more cost-effective alternatives to PSG are in high demand.

Consumer sleep-tracking technologies are rapidly advancing to meet increasing demand, resulting in novel tools and enhanced algorithms [1,2,3,4,6,26]. Wearable sleep technologies typically consist of wrist-worn or finger-worn sensors that monitor movement, cardiac activity, and even temperature [2,11,26]. Non-wearable sleep technologies, called nearables, typically measure physical and cardiac activity from the bedside or under a mattress without direct physical contact to the individual [2,26]. Nearables provide a potential solution for sleep measurement in populations sensitive to or unwilling to wear a device [27]. Compared to research-grade actigraphy, which relies on both participant and researcher input in the form of sleep diaries and hand scoring, wearable and nearable consumer devices are typically less demanding and measure a wider array of physiological signals [2,6,11,12,13,15,26]. These devices are also already in widespread public use and offer a promising opportunity to help improve sleep-related interventions and health. However, whether new consumer sleep-tracking technologies measure sleep reliably and accurately enough to be used as alternatives to PSG and research-grade actigraphy remains unclear and under evaluated.

In recent years, consumer devices have been able to report measures of sleep more accurately [3,4,28]. Devices that use accelerometry alone are able to accurately estimate sleep versus wake and tend to have the lowest staging accuracy at around 65% [29,30]. Devices with combined accelerometry with photoplethysmography have been able to achieve classification accuracies between 65 and 75% for sleep staging and 90% for classifying sleep and wake but tend to overestimate deep sleep and underestimate other sleep stages [4,31]. Pressure and radar-based systems have been demonstrated to achieve up to 80% accuracy at detecting light sleep and deep sleep [4]. Despite these tendencies, directly comparing device performance is difficult due to different, and often incomplete, evaluation and reporting methods and comparisons between participants [3,10,32]. Furthermore, new devices and algorithms are released nearly every month, and there is a critical lack of studies directly comparing sleep staging performance of different devices using a within-subject approach [32].

This study evaluated the accuracy of sleep measurement in five consumer sleep-tracking technologies and research-grade actigraphy against PSG, using a within-subject and standardized evaluation approach [10]. We were particularly interested in the performance of nearable devices (under mattress) and wearable devices with multiple measures (e.g., actigraphy, heart rate, SpO_2_) in comparison to research-grade actigraphy and PSG. Device measured total sleep time, wake after sleep onset, light sleep, deep sleep, and REM sleep were compared to their corresponding reference measures derived using consensus-scored PSG records. Such findings are important for identifying the clinical and research useability of these devices.

## 2. Materials and Methods

### 2.1. Participants

Participants consisted of 53 healthy young adults (31 female, 22 male) aged 18–30 years (M = 22.5, SD = 3.5). Participants were asked to wear the devices as part of other larger ongoing studies where participants slept in the lab with PSG. As part of the procedure of the ongoing studies, participants were excluded for self-reported cardiac, neurological, psychological, or sleep disorders (e.g., restless leg syndrome, insomnia, narcolepsy), abnormal sleep patterns (i.e., greater than 3 naps per week, fewer than 6 h of sleep per night, habitual bedtime after 2 a.m.), medication or supplements affecting sleep (e.g., Ambien, Lunesta, St. John’s Wort, melatonin, over-the-counter sleep supplements), and excessive caffeine or alcohol consumption (i.e., greater than fourteen 12 oz drinks per week). Participants were instructed to get quality sleep the night before experimental sessions and to abstain from caffeine and alcohol consumption on experimental days and prior nights. Information about sleep the night before and pre-testing criteria were verified by self-reports. All participants were provided monetary compensation for completing experimental procedures, and all procedures were approved by Institutional Review Board of the University of Massachusetts.

### 2.2. Surveys

Questionnaires assessed demographics, body mass index, and the presence of sleep disorders. The Morning–Eveningness Questionnaire (range (16–86), >59 = “morning types”) assessed participants’ chronotype [33]. The Pittsburg Sleep Quality Index (range (0–21), >5 = “Poor” sleeper) was used to assess habitual sleep over the last 30 days [34]. Self-reported information about participants’ sleep (e.g., sleep and wake onset, estimated sleep duration, awakenings) was collected using an in-house sleep diary.

### 2.3. Sleep-Tracking Devices

The Actiwatch Spectrum Plus (Phillips Respironics, Bend, OR, USA) is a research-grade activity-monitoring wristwatch. The devices were configured to collect data in 15 s epochs. Participants were instructed to press a button on the device to provide an event marker indicating when they went to bed after the lights were out, and when they woke up. Actigraphy data was downloaded with Actiware software 6.3.0 and scored for sleep or wake, or excluded if off-wrist, using 15 s epochs. Next, sleep diary entries and event markers were used to check the auto-scored rescored records for discrepancies with the sleep diaries and button presses. If there was no entry in the sleep diary or an event marker, the first 3 consecutive minutes of sleep defined sleep onset and the last 5 consecutive minutes of sleep defined wake onset [35].

We employed five commercial sleep-tracking devices. Cost and other information about the devices are outlined below in Table 1. The Fitbit Inspire HR (Fitbit Inc., San Francisco, CA, USA), Fitbit Versa 2 (Fitbit Inc. San Francisco, CA, USA), and Garmin Vivosmart 4 (Garmin Ltd., Lenexa, KA, USA) are smartwatches that pair with a dedicated smartphone application using Bluetooth. All 3 watches measure activity, sleep, and heart rate. The Fitbit Versa 2 and Garmin Vivosmart 4 have additional SpO_2_ monitors. The Oura Ring Gen 2 (Oura Inc., Oulu, Finland) is a smart ring that pairs with a dedicated smartphone application using Bluetooth and reports estimated measures of activity, sleep, and heart rate. We also assessed a nearable, the Withings Sleep Tracking Mat (Withings Inc., Issy-les-Moulineaux, France). This mat is placed under a mattress and paired with a smartphone application using Bluetooth.

All devices report estimated measures of total sleep time, wake after sleep onset, light sleep, deep sleep, and REM sleep for detected sleep intervals via the corresponding applications. For each device, the corresponding applications were used to extract summary measures. Epoch by epoch data was not available for any of the devices. Time stamps were not synchronized between the iPhone and PSG data collection computer. Therefore, only summary measures were assessed.

### 2.4. Polysomnography

Polysomnography for 33 participants was acquired using a 32-channel cap (Easycap, Herrsching, Germany) and a Bluetooth-compatible LiveAmp amplifier (Brain Products GmbH, Gilching, Germany). The EasyCap montage consisted of 28 EEG electrodes placed at 10–10 and intermediary locations, 2 electrooculogram (EOG) electrodes placed beside the eyes, and 2 electromyogram (EMG) electrodes placed over the zygomatic major and mylohyoid muscles. Data were recorded with a bandpass filter between RAW-250 Hz and were digitized at 500 Hz using BrainVision Recorder (Brain Products GmbH, Gilching, Germany). Scalp impedances were reduced below 10 kΩ.

For the remaining 20 participants, PSG was acquired using a custom 129-channel cap (Easycap, Herrsching, Germany) and BrainAmp MR plus amplifiers (Brain Products GmbH, Gilching, Germany). The PSG montage consisted of 123 scalp EEG electrodes placed at 10–10 and intermediary locations, 4 EOG electrodes placed beside and below the eyes, and 2 EMG electrodes placed over the zygomatic major and mylohyoid muscles. Data were recorded using a hardware bandpass filter between RAW-1000 Hz and digitized at 2000 Hz using BrainVision Recorder (Brain Products GmbH, Gilching, Germany). Scalp impedances were reduced below 10 kΩ using high-chloride abrasive gel.

### 2.5. Procedure

Data were collected in the Sleep Lab at The University of Massachusetts, Amherst. The phone applications for the Garmin Vivosmart 4, Withings Sleep Mat, and the Oura Ring were downloaded to an Apple iPhone 8. For each device, one account was created. Then, the same account for each device was used to collect data from all participants. Demographic information was updated for each participant prior to data collection. The Fitbit application is limited to one account per device, and only one account can log into the phone at a time. Therefore, one account was created for the Fitbit Versa, and another was created for the Fitbit HR. Data were harvested from both accounts by logging them in and out of the same application using the same phone following each study night.

While participating in other studies, participants were given information about the current study and asked whether they would be interested in wearing the devices alongside their already scheduled night PSG recording. All participants arrived at the lab between 8 and 11 p.m. to complete surveys and cognitive memory tasks associated with the studies. After the tasks, participants were given up to a 30 min break to prepare for overnight sleep in the lab (e.g., brush teeth, use restroom, change clothes). The Fitbit Versa and Garmin Vivosmart were applied to one wrist, and the Fitbit HR and Actiwatch Spectrum were placed on the other wrist. The Oura Ring was applied to whichever finger it fit on the best. Wrist location (proximal vs. distal) and arm placement (left vs. right) were counterbalanced across participants. The Withings Sleep Tracking Mat was placed under the mattress (full size, interspring mattress, Gold Medal Sleep Products) according to product specifications. Next, PSG was applied, and participants were instructed to sleep. Participants were in bed with PSG by between 9 p.m. and 12 a.m. In the morning, the PSG and devices were removed, participants completed any tasks required from the original studies, and then they were compensated and debriefed about the aims of this study.

### 2.6. Analysis

Sleep stages comprising NREM1, NREM2, NREM3, and REM were manually scored in 30 s epochs by two trained sleep researchers according to standard American Academy of Sleep Medicine criteria [18]. After each record was independently scored by each reviewer, discrepant epochs were identified. Each discrepant epoch was then individually rescored by another scorer to provide a consensus score. Rescored records with consensus scored epochs replacing discrepant scores were then used to calculate summary statistics for total sleep time, wake, NREM1, NREM2, NREM3, and REM sleep. Based on the prior literature, we assumed that NREM1 and NREM2 were both considered “light” sleep, and NREM3 was considered “deep” sleep [36]. Thus, reported data combined NREM1 and NREM2.

Absolute bias, Bland–Altman plots, intra-class correlation coefficients, and mean absolute percent error were used to evaluate whether the observed summary sleep measures recorded by each device differed from the corresponding measure recorded by PSG. Bias was assessed to determine how estimates differed as a function of the amount of each measurement. Plots of the mean difference and 95% limits of agreement were generated using recent standardized guidelines for the evaluation of sleep tracking [10]. Briefly, plots were examined for heteroscedasticity and proportional bias. Where heteroscedasticity and/or proportional bias were present, the bias and 95% limits of agreement (LOAs) were adjusted by log-transforming the data. When data were homoscedastic, LOAs were computed as bias ± 1.96 SD of the differences. When heteroscedasticity was detected, LOAs were modeled as a function of the size of the measurement [10].

Intra-class correlation coefficient values were calculated using an ICC (2,1) single-measurement, absolute-agreement, 2-way mixed-effects model and were classified as poor (<0.50), moderate (0.50–0.75), good (0.75–0.90), or excellent (>0.90) based on established guidelines [37]. Mean absolute percentage error values were calculated to indicate the relative measurement error of each consumer sleep-tracking device compared to PSG for all sleep measurements. Mean absolute percentage error was calculated as the absolute difference between the devices and PSG measures divided by the measured PSG and multiplied by 100 (e.g., [(Device TST − PSG TST)]/PSG TST × 100) [38,39]. All statistical analyses were performed using R [40].

## 3. Results

### 3.1. Sample Demographics

The sample demographics are presented in Table 2. Participants’ BMI ranges were in the healthy range. The MEQ indicated the participants were, on average, intermediate chronotypes with poor habitual sleep quality based on the PSQI scores (Table 3). Three participants slept less than 4.5 h overnight in the lab and were therefore excluded from further analyses. Sleep architecture of the final sample are in Table 4.

### 3.2. Device Failures

All devices exhibited software or user errors (Table 5). The Fitbit Inspire had three failures; two failures were related to poor fit obscuring a proper heart rate recording, and one was related to data synchronization. The Fitbit Versa had the most failures: nine were related to data synchronization and one was related to poor fit obscuring a proper heart rate recording. The Garmin Vivosmart had two errors related to data synchronization failures and one related to poor fit. For the Oura Ring, three errors were due to data synchronization failures, and one was due to low battery. For the Withings Mat, all six errors were due to data synchronization failures. The Actiwatch reported being off-wrist for more than 75% of the night seven times due to poor fit and failed to record one time because of an issue with programming of the device.

### 3.3. Total Sleep Time

Compared with PSG, the Phillips Actiwatch, Garmin Vivosmart, and Withings Mat overestimated total sleep time, while the Fitbit Inspire, Fitbit Versa, and Oura Ring tended to underestimate total sleep time. None of the devices had proportional bias for estimates of total sleep time. For all devices, heteroscedasticity manifested as the limits of agreement around their biases being wider for nights with longer total sleep times than nights with shorter total sleep times. The agreement between the devices and PSG-reported total sleep time was good for the Fitbit Inspire, Fitbit Versa, Withings Sleep Mat, and Oura Ring, moderate for the Phillips Actiwatch and poor for the Garmin Vivosmart (Figure 1 and Table 6).

### 3.4. Wake after Sleep Onset

Compared with PSG, all devices tended to overestimate wake after sleep onset. All devices tended to overestimate nights with shorter wake times after sleep onset times and underestimated nights with longer wake times after sleep onset times. For all devices, heteroscedasticity manifested as the limits of agreement being wider for nights with longer wake times after sleep onset times than for nights with shorter wake times after sleep onset times. The agreement between the devices and PSG-reported wake after sleep onset was poor for all devices (Figure 2 and Table 7).

### 3.5. Light Sleep

Compared with PSG-measured light sleep (NREM1 + NREM2), the Fitbit Inspire and Fitbit Versa tended to slightly overestimate light sleep. Bias was not proportional, and the limits of agreement were wider for nights with longer light sleep times than for nights with shorter light sleep times. The Garmin Vivosmart and Withings Mat had proportional bias with heteroscedasticity such that light sleep tended to be overestimated at lower light sleep times and underestimated at higher light sleep times. Limits of agreement were wider for nights with longer light sleep times than for nights with shorter light sleep times. The Oura Ring tended to underestimate light sleep, and limits of agreement were wider for longer light sleep times than short ones. The agreement between devices and PSG-reported wake after sleep onset was poor for all devices (Figure 3 and Table 8).

### 3.6. Deep Sleep

The Fitbit Inspire, Fitbit Versa, and Oura Ring tended to overestimate deep sleep at lower deep sleep times and tended to underestimate higher deep sleep times compared to PSG-measured NREM3. Limits of agreement were wider for nights with longer deep sleep times than for nights with shorter deep sleep times. The Garmin Vivosmart and Withings Sleep Mat tended to overestimate deep sleep, and limits of agreement were wider for longer deep sleep times than shorter ones. The agreement between devices and PSG-reported wake after sleep onset was poor for all devices (Figure 4 and Table 9).

### 3.7. REM Sleep

All of the devices tended to overestimate REM sleep. None of the devices had proportional bias. Limits of agreement were wider for nights with longer compared to shorter REM times. The agreement between devices and PSG-reported REM sleep was moderate for the Fitbit Inspire and Fitbit Versa, and was poor for all other devices (Figure 5 and Table 10).

### 3.8. Mean Absolute Percent Error

The mean absolute percent error for total sleep time was the lowest and ranged from 5.3% (Fitbit Versa) to 14.3% (Garmin Vivosmart), indicating acceptable accuracy for all devices except the Oura Ring. The mean absolute percent error for wake after sleep onset was the highest and ranged from 59.29% (Phillips Actiwatch) to 138.5% (Fitbit Versa), indicating very low accuracy across all devices. The mean absolute percentage errors for light sleep were between 19.4% (Garmin Vivosmart) and 27.0% (Oura Ring), indicating low but acceptable accuracy. The mean absolute percent errors for deep sleep and REM sleep were all above 20%, indicating low accuracy across all devices (Figure 6).

## 4. Discussion

Here, we compared sleep measurements from commercial wearable and nearable sleep-tracking devices and research-grade actigraphy relative to gold-standard PSG. Two core findings emerged from this research. All but one of the consumer sleep-tracking technologies were able to detect total sleep time with comparable accuracy to research-grade actigraphy. Second, the accuracy of device-measured sleep stages depends on many factors, including the sleep stage, the device, time spent in that sleep stage, and the individual. Taken together, these findings support a growing body of literature that consumer sleep-tracking devices are a potential cost-effective, convenient, and accurate alternative sleep measurement tool.

All devices assessed here detected total sleep time with similar accuracy to research-grade actigraphy, without requiring any programming, software, button presses, sleep diaries, or a researcher to score records. Consistent with a growing number of recent studies, we find that newer-generation wearable and nearable devices are capable of accurately estimating sleep time within clinically acceptable levels [5,11,36,41]. Also, the reliability and convenience of consumer sleep-tracking technologies was demonstrated to be better than research-grade actigraphy [5,11]. Our results demonstrate that all devices, except the Garmin Vivosmart, were able to estimate total sleep time with biases, reliability, and accuracy comparable to research-grade actigraphy (Figure 7 and Figure 8). All of the devices also cost less than research-grade actigraphy.

For wake detection, interestingly, all of the tested devices and research-grade actigraphy overestimated nights with shorter wake after sleep onset times. All of the devices also underestimated nights with longer wake after sleep onset times. Moreover, all of the devices had low accuracy and a higher error for wake after sleep onset than any other sleep measurement. Due to the difficulty with detecting when someone is awake but not moving, devices and algorithms that rely heavily on actigraphy were reported to largely underestimate wake after sleep onset [10,42,43]. In newer-generation wearable and nearable devices, accurate heart rate and heart rate variability measures were widely incorporated into sleep and wake detection algorithms [2,3,4,11]. These results contribute to a growing body of evidence supporting that many new consumer sleep-tracking devices with heart rate sensors can track sleep and wake as accurately as or better than actigraphy [5,36,41]. However, improvements are required before any of these devices can replace the accuracy of PSG for detecting when someone is awake.

It remains unclear whether and how different sensors and physiological signals are being incorporated for sleep staging across consumer sleep-tracking technologies [10,12]. In this analysis, bias in sleep staging estimates varied greatly across individuals, devices, stages, and stage length. Further, the accuracy of all sleep measurements compared to PSG was poor. Similar to previous studies, discerning light sleep from deep sleep was especially challenging for sleep-tracking devices [5,31,36,41,44,45,46,47,48,49,50,51]. The Fitbit Inspire, Fitbit Versa, and Oura Ring estimated light sleep with less bias than deep sleep. Conversely, the Withings Mat and Garmin Vivosmart estimated deep sleep with less bias than light sleep. Across devices, estimates for REM sleep were less biased than estimates of light and deep sleep. Despite low absolute biases, it should be noted that some estimates were still up to 250 min different than their PSG derived measures. Moreover, overestimations and underestimations were dependent on the length of the estimate for some stages but not others across all of the sleep-tracking technologies tested. Taken together, the results of this research suggest that proportional bias patterns in consumer sleep-tracking technologies are prevalent and could have important implications for the overall accuracy of devices.

Although the sleep staging performance of consumer sleep-tracking devices is increasing, compared to research-grade actigraphy, the mechanisms and specific causes of bias and variability are less transparent. To classify sleep stages, wearable and nearable devices implement a wide array of proprietary sensors and algorithms, whereas the algorithms and data from the Actiwatch and other research-grade devices are open to and tested by researchers. Adding to the obscurity of consumer-grade devices, it remains difficult or expensive to gather the necessary data to properly evaluate the accuracy of consumer sleep staging devices and their sensors at a full resolution. Some devices provide access to all of the data collected from device sensors, but obtaining access to the data requires advanced coding knowledge, the ability to utilize application programming interfaces, or funding to pay the company or a third party to develop custom software. Research-grade actigraphy devices provide full access to the data, dedicated software, and open-source scoring algorithms. In this study, epoch-by-epoch data were not accessed for any of the devices due to technological constraints. Therefore, the specificity and accuracy at which any of the tested devices detected sleep staging remains unclear. Future work should prioritize the development of free open-source tools designed to facilitate the extraction of epoch-by-epoch data from consumer sleep-tracking devices. Facilitating a greater access to granular data will be essential to hasten the evaluation of specific causes of bias and variability, such as race, which remained to be determined [52].

## 5. Limitations

There are limitations to consider in the interpretation of these results. First, devices were worn for one night. The accuracy of measurements was demonstrated to vary due to a variety of factors including night to night differences [27,53]. Second, commercial devices are rapidly evolving, and validity may be improved in newer models. However, in the past, upgrades to the same device have not improved the sensitivity, specificity, and accuracy for any sleep measure [48,54]. Finally, our sample was not sufficiently diverse enough to know the generalizability of these findings across racial and ethnic groups. Photoplethysmography, which provides a measure of heart rate (which is used in sleep estimation) by shining light through the skin to detect blood flow changes is used in many newer wearable devices (Versa, Inspire, Vivofit, Oura) and is not as accurate in individuals with darker skin tones [52,55,56,57,58]. Thus, it is critical for future studies to include more diverse samples and separately validate performance in individuals with dark skin tones.

## 6. Conclusions

The results of this study support a growing body of literature that consumer sleep-tracking devices are a potential widely available, cost-effective, and accurate alternative sleep measurement tool. According to a recent national survey, nearly 30% of US adults use a wearable device on a daily basis, and the majority of those users would be willing to share their information with health care providers [59]. Wearable devices have been demonstrated to facilitate the success of sleep interventions and could help improve health outcomes for a large portion of the population [60]. Consistent with a growing number of recent studies, all but one of the newer-generation wearable and nearable devices tested in this study were capable of accurately estimating total sleep within clinically acceptable levels [5,36]. Despite their prevalence and potential utility, only 5% of consumer sleep-tracking technologies have been formally evaluated [28,61]. Many previous evaluation approaches have not examined whether accuracy differs across shorter compared to longer staging estimates [10,12,28]. Here, we demonstrated prevalent proportional bias patterns in consumer sleep-tracking technologies that could have important implications for their overall accuracy. Despite their prevalence, it remains unclear whether and how different sensors, algorithms, and physiological signals are being incorporated across consumer sleep-tracking technologies. Greater access to granular data and future studies to further investigate the implications of proportional bias across devices that use different sensors and physiological measures are warranted. While the performance of some devices is strong, we also note the need for further improvements to the reliability and accuracy of consumer devices and research-grade actigraphy.

## Figures and Tables

**Figure 1 sensors-24-00635-f001:**
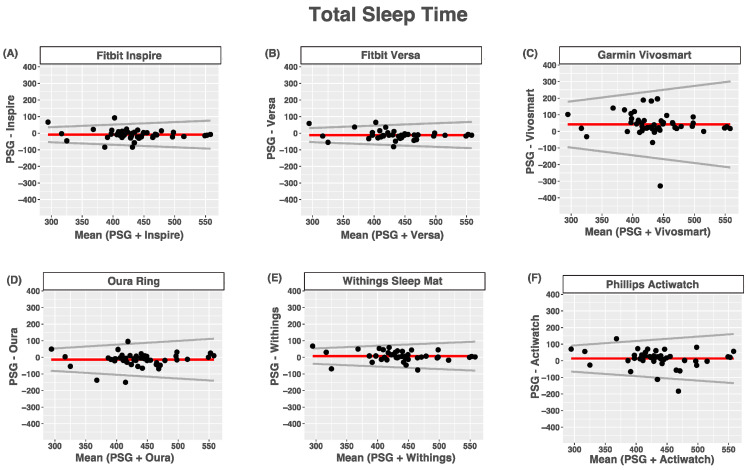
Bland–Altmann plots depicting device-measured and PSG-measured agreement for total sleep time. (**A**) Fitbit Inspire; (**B**) Fitbit Versa; (**C**) Garmin Vivosmart 4; (**D**) Oura Ring; (**E**) Withings Sleep Mat; (**F**) Phillips Actiwatch. Data are shown in minutes. Red lines = mean bias, grey lines = upper and lower limits of agreement, dots = individuals.

**Figure 2 sensors-24-00635-f002:**
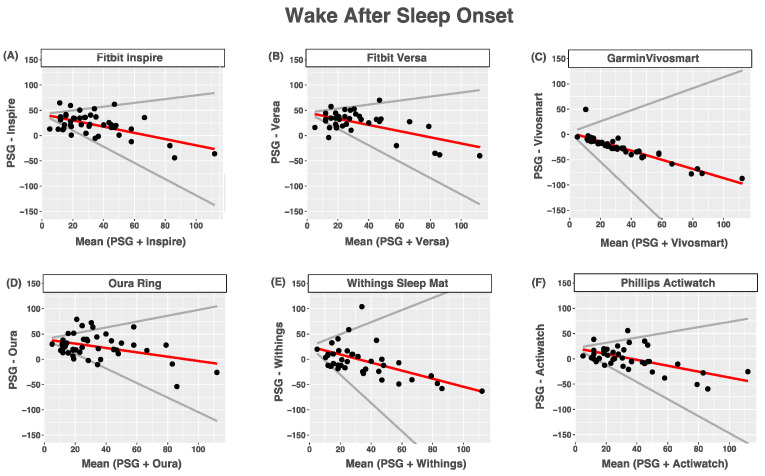
Bland–Altmann plots depicting device-measured and PSG-measured agreement for wake after sleep onset (**A**) Fitbit Inspire; (**B**) Fitbit Versa; (**C**) Garmin Vivosmart 4; (**D**) Oura Ring; (**E**) Withings Sleep Mat; (**F**) Phillips Actiwatch. Data are shown in minutes. Red lines = mean bias, grey lines = upper and lower limits of agreement, dots = individuals.

**Figure 3 sensors-24-00635-f003:**
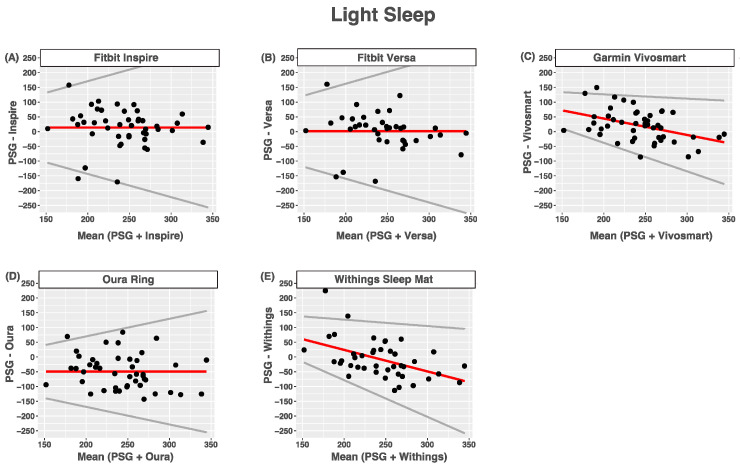
Bland-Altmann plots depicting device-measured and PSG-measured agreement for light sleep. (**A**) Fitbit Inspire; (**B**) Fitbit Versa; (**C**) Garmin Vivosmart; (**D**) Oura Ring; (**E**) Withings Sleep Mat. Data are shown in minutes. Red lines = mean bias, grey lines = upper and lower limits of agreement, dots = individuals.

**Figure 4 sensors-24-00635-f004:**
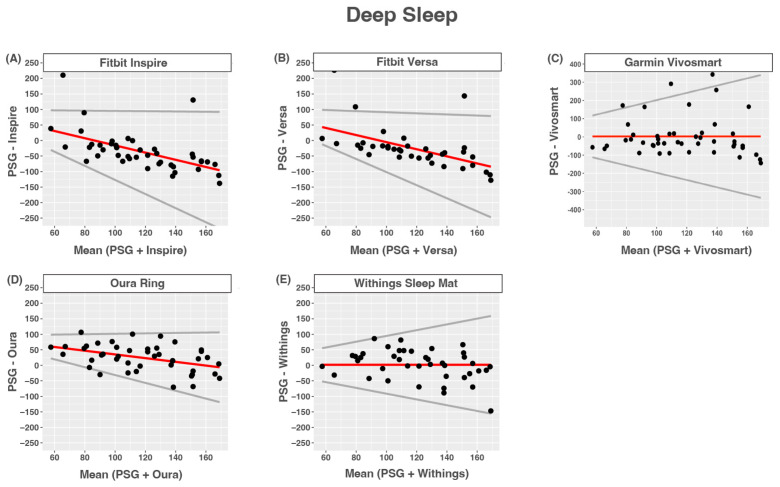
Bland–Altmann plots depicting device-measured and PSG-measured agreement for deep sleep. (**A**) Fitbit Inspire; (**B**) Fitbit Versa; (**C**) Garmin Vivosmart 4; (**D**) Oura Ring; (**E**) Withings Sleep Mat. Data are shown in minutes. Red lines = mean bias, grey lines = upper and lower limits of agreement, dots = individuals.

**Figure 5 sensors-24-00635-f005:**
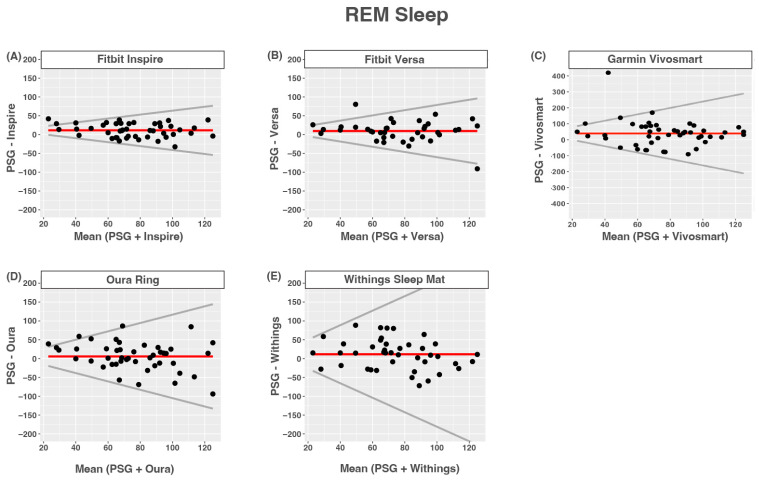
Bland–Altmann plots depicting device-measured and PSG-measured agreement for REM sleep. (**A**) Fitbit Inspire; (**B**) Fitbit Versa; (**C**) Garmin Vivosmart 4; (**D**) Oura Ring; (**E**) Withings Sleep Mat. Data are shown in minutes. Red lines = mean bias, grey lines = upper and lower limits of agreement, dots = individuals.

**Figure 6 sensors-24-00635-f006:**
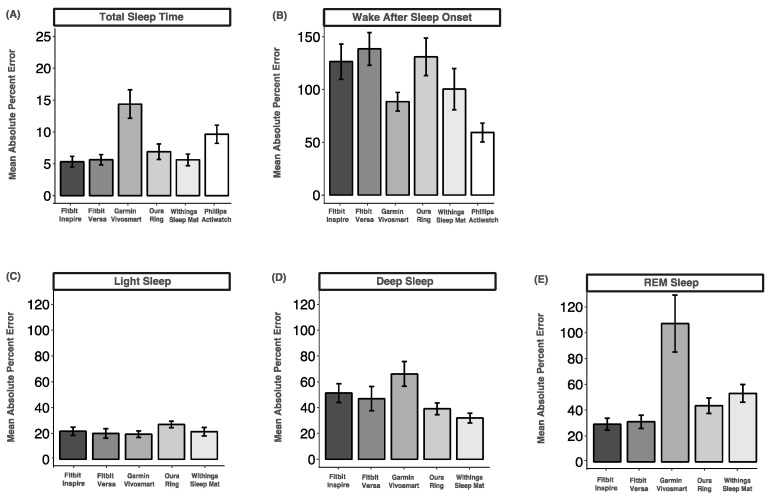
Mean absolute percent error comparisons between device-measured and PSG-measured sleep variables. (**A**) Total sleep time; (**B**) wake after sleep onset; (**C**) light sleep; (**D**) deep sleep; (**E**) REM sleep. Data are shown as mean ± standard error.

**Figure 7 sensors-24-00635-f007:**
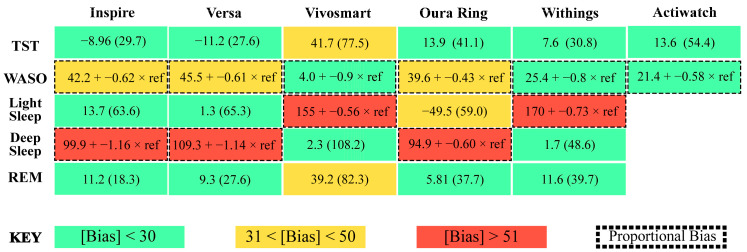
Summary figure depicting absolute bias in sleep measurements for each device. TST = total sleep time, WASO = wake after sleep onset, green boxes = absolute bias < 30, yellow boxes = absolute bias between 31 and 50, red boxes = absolute bias > 51, boxes with dotted outlined = proportional bias.

**Figure 8 sensors-24-00635-f008:**
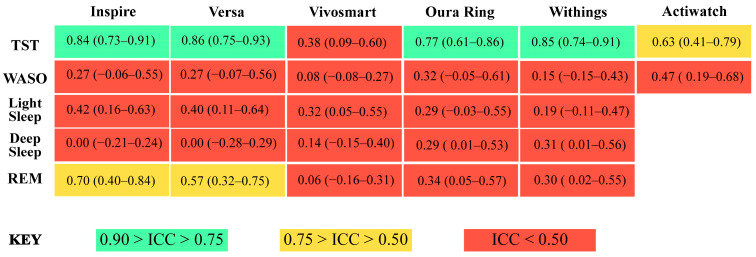
Summary figure depicting intra-class correlation (ICC) values and 95% confidence intervals of agreement between device- and PSG-measured sleep variables. TST = total sleep time, WASO = wake after sleep onset, light green boxes = 0.90 > ICC > 0.75; good accuracy, yellow boxes = 0.75 > ICC > 0.50; moderate accuracy, red boxes = ICC < 0.50; poor accuracy.

**Table 1 sensors-24-00635-t001:** Information about cost, location, and available measurements for each device.

	Cost	Location	Measurements
Fitbit Inspire HR	USD 97	Wrist	Activity, Sleep, Heart Rate
Fitbit Versa 2	USD 200	Wrist	Activity, Sleep, Heart Rate, SpO_2_
Garmin Vivosmart	USD 110	Wrist	Activity, Sleep, Heart Rate, SpO_2_
Oura Ring	USD 300	Finger	Activity, Sleep, Heart Rate
Withings	USD 80	Under Mattress	Activity, Sleep, Heart Rate
Actiwatch	USD 800	Wrist	Activity, Sleep

**Table 2 sensors-24-00635-t002:** Demographics of the participants included in this study.

	Total	F:M	White	Asian	Hispanic	Black	DNR
N	53	31:22	29	11	7	3	3

F:M = females/males, DNR = did not report.

**Table 3 sensors-24-00635-t003:** Summary of participant characteristics.

	Age (Years)	Height (m)	Weight (kg)	BMI (kg/m^2^)	MEQ *	PSQI
Mean (SD)	22.5 (3.5)	1.7 (0.10)	69.4 (16.6)	23.7 (5.0)	45.4 (8.4)	5.9 (2.8)
Median	22.0	1.7	65.8	22.7	45.5	6.0
Range	18–34	1.5–1.9	49.9–131	16.9–44.4	26–64	1–17

SD = standard deviation, BMI = body mass index, MEQ = Morning–Eveningness Questionnaire, PSQI = Pittsburgh Sleep Quality Index. * Scores unavailable for 3 participants.

**Table 4 sensors-24-00635-t004:** Summary statistics for sleep stages calculated by polysomnography in minutes.

	TST	WASO	NREM1	NREM2	NREM3	REM
Mean (SD)	431 (54.7)	33.1 (22.6)	15.7 (10.0)	226 (38.4)	113 (35.2)	76.4 (25.2)

TST = total sleep time, WASO = wake after sleep onset.

**Table 5 sensors-24-00635-t005:** Device failures.

Device	% of Data Lost	N Errors Total	Data Synch	Poor Fit	Low Battery
Fitbit Inspire	5.6%	3	1	2	0
Fitbit Versa	18.8%	10	9	1	0
Vivosmart	5.6%	3	2	1	0
Oura Ring	7.5%	4	3	0	1
Withings	11.3%	6	6	0	0
Actiwatch	15.1%	8	1	7	0

Data Synch = data synchronization or programming failures.

**Table 6 sensors-24-00635-t006:** Agreement between devices and PSG for total sleep time.

	Device (min)	PSG (min)	Bias	LOA	ICC
Inspire	424.5 (66.4)	433.5 (52.7)	−8.96 (29.7)	0.15	0.84 (0.73–0.91)
Versa	423.7 (57.7)	434.9 (56.6)	−11.2 (27.6)	0.14	0.86 (0.75–0.93)
Vivosmart	476.6 (87.9)	435.0 (53.5)	41.7 (77.5)	0.46	0.38 (0.09–0.60)
Oura Ring	420.7 (70.6)	434.6 (54.1)	−13.9 (41.1)	0.23	0.77 (0.61–0.86)
Withings	443.6 (58.9)	436.0 (55.9)	7.6 (30.8)	0.16	0.85 (0.74–0.91)
Actiwatch	449.0 (71.1)	435.4 (56.7)	13.6 (54.4)	0.26	0.63 (0.41–0.79)

Bias = absolute bias and standard deviation. LOA = limits of agreement (Bias ± ref × value), ICC = intraclass correlation coefficient and 95% confidence intervals.

**Table 7 sensors-24-00635-t007:** Bias and limits of agreement between device- and PSG-measured wake after sleep onset.

	Device (min)	PSG (min)	Bias	LOA	ICC
Inspire	54.8 (20.1)	32.7 (22.2)	42.2 + −0.62 × ref	0.99	0.27 (−0.06–0.55)
Versa	58.5 (21.9)	33.5 (24.3)	45.5 + −0.61 × ref	1.01	0.27 (−0.07–0.56)
Vivosmart	7.2 (10.1)	33.1 (22.9)	4.0 + −0.9 × ref	2.00	0.08 (−0.08–0.27)
Oura Ring	58.8 (26.6)	33.6 (23.2)	39.6 + −0.43 × ref	1.01	0.32 (−0.05–0.61)
Withings	32.2 (33.2)	33.6 (23.9)	25.4 + −0.8 × ref	2.00	0.15 (−0.15–0.43)
Actiwatch	35.9 (21.1)	34.7 (23.5)	21.4 + −0.58 × ref	1.10	0.47 (0.19–0.68)

Bias = absolute bias and standard deviation. LOA = limits of agreement (Bias ± ref × value), ICC = intraclass correlation coefficient and 95% confidence intervals.

**Table 8 sensors-24-00635-t008:** Bias and limits of agreement between device- and PSG-measured light sleep.

	Device (min)	PSG (min)	Bias	LOA	ICC
Inspire	256.8 (73.0)	243.0 (42.2)	13.7 (63.6)	0.79	0.42 (0.16–0.63)
Versa	246.2 (72.2)	244.9 (43.1)	1.3 (65.3)	0.80	0.40 (0.11–0.64)
Vivosmart	263.3 (51.7)	243.0 (41.3)	155 + −0.56 × ref	0.41	0.32 (0.05–0.55)
Oura Ring	193.5 (64.3)	243.0 (42.2)	−49.5 (59.0)	0.60	0.29 (−0.03–0.55)
Withings	235.9 (58.5)	243.6 (42.7)	170 + −0.73 × ref	0.51	0.19 (−0.11–0.47)

Bias = absolute bias and standard deviation. LOA = limits of agreement (Bias ± ref × value), ICC = intraclass correlation coefficient and 95% confidence intervals.

**Table 9 sensors-24-00635-t009:** Bias and limits of agreement between device- and PSG-measured deep sleep.

	Device (min)	PSG (min)	Bias	LOA	ICC
Inspire	81.5 (52.3)	117.7 (30.4)	99.9 + −1.16 × ref	1.10	0.00 (−0.21–0.24)
Versa	92.2 (56.3)	117.2 (30.8)	109.3 + −1.14 × ref	0.96	0.00 (−0.28–0.29)
Vivosmart	120.3(110)	117.9 (30.3)	2.3 (108.2)	1.99	0.14 (−0.15–0.40)
Oura Ring	142.5(40.5)	118.5 (30.9)	94.9 + −0.6 × ref	0.67	0.29 (0.01–0.53)
Withings	122.8(48.1)	121.1 (30.2)	1.7 (48.6)	0.93	0.31 (0.01–0.56)

Bias = absolute bias and standard deviation. LOA = limits of agreement (Bias ± ref × value), ICC = intraclass correlation coefficient and 95% confidence intervals.

**Table 10 sensors-24-00635-t010:** Bias and limits of agreement between device- and PSG-measured REM sleep.

	Device (min)	PSG (min)	Bias	LOA	ICC
Inspire	86.8 (27.4)	75.6 (24.2)	11.2 (18.3)	0.52	0.70 (0.40–0.84)
Versa	85.8 (34.6)	76.5 (26.6)	9.3 (27.6)	0.69	0.57 (0.32–0.75)
Vivosmart	115.6 (81.8)	76.3 (25.0)	39.2 (82.3)	2.00	0.06 (−0.16–0.31)
Oura Ring	81.8 (38.9)	75.9 (25.4)	5.81 (37.7)	1.11	0.34 (0.05–0.57)
Withings	85.8 (41.0)	74.2 (24.7)	11.6 (39.7)	1.93	0.30 (0.02–0.55)

Bias = absolute bias and standard deviation. LOA = limits of agreement (Bias ± ref × value), ICC = intraclass correlation coefficient and 95% confidence intervals.

## Data Availability

The data presented in this study are available upon reasonable request from the corresponding author.

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
