# Peer review of "Evaluating Accuracy in Five Commercial Sleep-Tracking Devices Compared to Research-Grade Actigraphy and Polysomnography"

_sensors, 2024, doi:10.3390/s24020635_

Round 1

Reviewer 1 Report

Comments and Suggestions for Authors

Authors compared five consumer grade sleep trackers and a research grade wrist worn actigraph to a gold standard PSG sleep examination to reveal that their metrological inaccuracy is in most cases a fair trade off for cost-effectiveness and wearers' comfort. I strongly support the approach resulting in widespreading of interest for quality seep and daily use of related wearables at home. What authors could do more is to prioritize sleep quality measures in regard to healthy lifestyle.

Few detailed minor remarks follow:

1. Besides the result quality is there any advantage of research-grade actigraph justifying its price?

2. Device failures reporting is an advantage of the manuscript, however, out of 52 people, the reliability of devices (including the research grade one) hardly excesses 80% 

3. In Table 3 units should be provided are height in inches and weight in pounds? The reader will prefer values in SI units.

4. In Table 6 units should be provided (minutes?) The same for figs 1 and 2 (Bland Altman plots) also in subsequent similar tables and plots.

5. In all subplots of Fig 2 distributions (gray by right edges of plots) seem to be shifted up relative to the dots in the plot. Please check also in Fig. 5c

6. I suggest providing formula for calculation to make sure the proper understanding of 'limits of agreement' in Tab. 6-10.

Author Response

1. What authors could do more is to prioritize sleep quality measures in regard to healthy lifestyle.

We have added language to section 1. Introduction (pg 2, lines 66-67) and section 5 (pg 15, lines 473-475) to reflect the potential for sleep tracking in regard to health lifestyle.

2. Besides the result quality is there any advantage of research-grade actigraph justifying its price?

Research grade actigraphy devices are more widely used, have been used for longer, provide greater access to both the data and scoring algorithms, and are more customizable for researchers. This has been clarified (pg 14, lines 448-459).

3. Device failures reporting is an advantage of the manuscript, however, out of 52 people, the reliability of devices (including the research grade one) hardly excesses 80% 

We have added a column clarifying the percent of data lost due to all errors in Table 5 (pg 6, lines 257-258). We have also elaborated that there is room for improvement for the reliability of both consumer and research grade sleep monitoring devices (pg 15, lines 487-489).

4. In Table 3 units should be provided are height in inches and weight in pounds? The reader will prefer values in SI units.

The units in Table 3 (pg 6, lines 241-244) have been labeled and converted to SI units.

5. In Table 6 units should be provided (minutes?) The same for figs 1 and 2 (Bland Altman plots) also in subsequent similar tables and plots.

The units in all tables and figures have been clarified.

6. In all subplots of Fig 2 distributions (gray by right edges of plots) seem to be shifted up relative to the dots in the plot. Please check also in Fig. 5c

The density distributions have been removed from all plots.

7. I suggest providing formula for calculation to make sure the proper understanding of 'limits of agreement' in Tab. 6-10.

The limits of agreement are calculated using the formula provided in the table caption. LOA = Bias ± ref x value. Table captions 6-10 have been edited to reflect this. Additional text has also been added to section 2.6 Analysis (pg 5, lines 218-220). 

Reviewer 2 Report

Comments and Suggestions for Authors

This is a well written manuscript. Below are my comments.

Introduction:

The introduction is very well written

Methodology:

The methodology needs significantly more information for replication.

1. Was this part of a larger study? Based on the procedures section, this might be?

2. How were participants recruited? How many volunteered to participate but did not meet inclusion/exclusion criteria?

3. Did you only monitor sleep for 1 night?

4. Did you track sleep the night before? How did you determine whether participants had met pre-testing criteria?

5. What time did participants arrive in lab? What time did they leave? Was this standardized or was this individualized based on their self-reported sleep patterns? (e.g. from the Morning-Eveningness survey or from PSQI)

6. Did you assess self-reported sleep quality the morning after?

7. Why did you not use MAE, MAPE and MSE to assess errors? What about ICCs between each device and PSG.

I believe these analyses will be extremely helpful in determining not only bias but error. Based on some of your plotted points on the Bland-Altman, the MSEs for some of these devices may be very high and might be truly revealing of the validity of the measurement of these devices. 

What you are measuring is just bias in the data, but you are not determining whether this is a valid tool or not.

Author Response

1. Was this part of a larger study? Based on the procedures section, this might be?

This study was an optional add-on to ongoing cognitive memory studies in our lab. While participating in other studies, participants were provided information about the current study, and asked to wear the devices in addition to their already scheduled polysomnographic recording. This has been clarified in section 2.1 (pg 2-3, 97-110) and 2.5 (pg 4, lines 183-186).

2. How were participants recruited? How many volunteered to participate but did not meet inclusion/exclusion criteria?

Participants were recruited according to the screening criteria outlined in section 2.1 (pg 2, lines 98-111). Everyone who volunteered to participate was eligible because they had been prescreened as part of the procedure of original studies that this one was added on to. This has been clarified in section 2.1 (pg 2-3, lines 98-100).

3. Did you only monitor sleep for 1 night?

Sleep was monitored for 1 night in all participants. This has been clarified in section 2.5 (pg 4, lines 183-186) and the abstract (line 16).

4. Did you track sleep the night before? How did you determine whether participants had met pre-testing criteria?

We did not track sleep the night before; rather, we relied on participant self-report. This has been clarified in section 2.1 (pg 3, lines 108-109).

5. What time did participants arrive in lab? What time did they leave? Was this standardized or was this individualized based on their self-reported sleep patterns? (e.g. from the Morning-Eveningness survey or from PSQI)

Participants arrived at the lab between 8 -11 pm as part of the ongoing procedures of the original studies that this study was added on to. All participants were in bed with PSG by between 9 pm-12 am. This has been clarified in section 2.5 (pg 5, lines 195-198).

6. Did you assess self-reported sleep quality the morning after?

We did not assess self-reported sleep quality, only bedtimes and wake times. While it would be interesting to compare subjective sleep quality and device reported metrics of sleep quality such as “sleep scores”, this was outside of the present scope.

7. Why did you not use MAE, MAPE and MSE to assess errors? What about ICCs between each device and PSG. I believe these analyses will be extremely helpful in determining not only bias but error. Based on some of your plotted points on the Bland-Altman, the MSEs for some of these devices may be very high and might be truly revealing of the validity of the measurement of these devices. What you are measuring is just bias in the data, but you are not determining whether this is a valid tool or not.

These are great considerations. Section 2.6 (pg 5, lines 210-231) has been updated to reflect the implementation of ICCs and MAPE to assess error. ICCs between each device and PSG, for each sleep variable, have been added to Tables 6-10 and Results (pg 6-10). The MAPE has also been calculated for each device and sleep measurement and incorporated into the Results with the addition of section 3.8 Mean Absolute Percent Error, and Figure 6 (pg 11-12, lines 371-378), and into the Discussion (pg 12-15).

Round 2

Reviewer 2 Report

Comments and Suggestions for Authors

I appreciate the authors incorporating my suggestions. The only thing that the manuscript is now missing is a limitations section. A limitation to keep in mind is that many of these devices use PPG to measure HR and HRV and your diverse population with various skin tones there may have been variability in the accuracy of those readings. It would be preferred if you collected multiple nights of data on these individuals to assess how these measures vary from day to day and whether they are dependent on skin tone.

Another very minor comment is that your demographic data is clearly skewed. I'd recommend reporting median and range.

Author Response

I appreciate the authors incorporating my suggestions. The only thing that the manuscript is now missing is a limitations section. A limitation to keep in mind is that many of these devices use PPG to measure HR and HRV and your diverse population with various skin tones there may have been variability in the accuracy of those readings. It would be preferred if you collected multiple nights of data on these individuals to assess how these measures vary from day to day and whether they are dependent on skin tone.

Section 5. Limitations has been added to the manuscript (pg 15, lines 467-481) and these points have been addressed.

Another very minor comment is that your demographic data is clearly skewed. I'd recommend reporting median and range.

The median and range have been added to Table 3 (pg 6, lines 242-244). Note also that this brought our attention to an error in the reported average height which is now corrected (thank you!).